# Effect of Planting Systems on the Physicochemical Properties and Bioactivities of Strawberry Polysaccharides

**DOI:** 10.3390/foods14020238

**Published:** 2025-01-14

**Authors:** Qiuqiu Zhang, Renshuai Huang, Guangjing Chen, Fen Guo, Yan Hu

**Affiliations:** College of Food Science and Engineering, Guiyang University, Guiyang 550005, China; zhangqq1248@163.com (Q.Z.); huangrenshuai@126.com (R.H.); gjchen1989@126.com (G.C.); guofen1104@163.com (F.G.)

**Keywords:** strawberry fruit polysaccharide, diffident planting, non-enzymatic glycosylation, hypoglycemic activity, rheological behavior

## Abstract

Suitable planting systems are critical for the physicochemical and bioactivities of strawberry (*Fragaria × ananassa* Duch.) polysaccharides (SPs). In this study, SPs were prepared through hot water extraction, and the differences in physicochemical characteristics and bioactivities between SPs derived from elevated matrix soilless planting strawberries (EP-SP) and those from and conventional soil planting strawberries (GP-SP) were investigated. A higher extraction yield was observed for EP-SP (5.88%) than for GP-SP (4.67%), and slightly higher values were measured for the average molecular weight (632.10 kDa vs. 611.88 kDa) and total sugar content (39.38% vs. 34.92%) in EP-SP. In contrast, a higher protein content (2.12% vs. 1.65%) and a more ordered molecular arrangement were exhibited by GP-SP. Monosaccharide composition analysis revealed that EP-SP contained higher levels of rhamnose (12.33%) and glucose (49.29%), whereas GP-SP was richer in galactose (11.06%) and galacturonic acid (19.12%). Thermal analysis indicated only minor differences in decomposition temperatures (approximately 225–226 °C) and thermal stability between the samples. However, GP-SP showed a higher enthalpy change (Δ*Hg* = 18.74 J/g) compared to EP-SP (13.93 J/g). Biological activity assays revealed that GP-SP generally exerted stronger non-enzymatic glycation inhibition at both early and final stages (IC_50_: 7.47 mg/mL vs. 7.82 mg/mL and 11.18 mg/mL vs. 11.87 mg/mL, respectively), whereas EP-SP was more effective against intermediate α-dicarbonyl compounds (maximum inhibition of 75.32%). Additionally, GP-SP exerted superior α-glucosidase inhibition (IC_50_ = 2.4583 mg/mL), in line with kinetic and fluorescence quenching analyses showing a higher enzyme–substrate complex binding affinity (*Kis* = 1.6682 mg/mL; *Ka* = 5.1352 × 10^5^ M^−1^). Rheological measurements demonstrated that EP-SP solutions exhibited a pronounced increase in apparent viscosity at higher concentrations (reaching 3477.30 mPa·s at 0.1 s^−1^ and 70 mg/mL) and a stronger shear-thinning behavior, while GP-SP showed a comparatively lower viscosity and lower network order. These findings suggest that different planting systems significantly affect both the molecular structures and functionalities of SPs, with GP-SP demonstrating enhanced hypoglycemic and anti-glycation properties. It is therefore recommended that suitable planting systems be selected to optimize the functionality of plant-derived polysaccharides for potential applications in the food and pharmaceutical industries.

## 1. Introduction

Plant polysaccharides are extensively utilized in food and various fields of human life [1]. Recently, significant attention has been directed toward their beneficial properties, including antioxidant, anticancer, antiviral, hypoglycemic, and immunomodulatory activities [2,3]. Their bioactivities have been shown to be influenced by monosaccharide composition, molecular weight, uronic acid and sulfate content, polymerization, branching degree, and polymer chain flexibility. Polysaccharides have been reported to function as natural inhibitors of digestive enzymes, safely and effectively delaying carbohydrate digestion. Additionally, adverse vascular effects in diabetic patients can be mitigated through the inhibition of advanced glycation end products (AGEs) [4], wherein reducing sugars nonenzymatically react with amino groups in proteins, nucleic acids, and lipids [5]. The accumulation of AGEs in blood and tissues has been associated with diseases such as cardiovascular disease, aging, and diabetes [6]. Although synthetic drugs are available to inhibit glycation, many pose toxicity or side effects with long-term use. Plant polysaccharides have been found to exhibit excellent anti-glycation activity [7], making dietary interventions involving polysaccharide-rich fruits or vegetables an effective strategy to reduce the risk of chronic diseases like type 2 diabetes. It has been reported that the presence of carboxylic acid and hydroxyl groups in the branched chains of polysaccharides facilitates the formation of strong hydrogen bonds with digestive enzymes, increasing exposure to active sites and inducing spatial configuration changes that partially inhibit enzyme activity [8]. Furthermore, polysaccharides rich in glucuronic acid with keto groups or galactose with hydroxyl groups have been confirmed to enhance enzyme binding, effectively inhibiting enzyme activity [9]. It is therefore crucial to investigate the physicochemical characteristics and bioactivities of polysaccharides to guide their applications in the food and pharmaceutical industries.

The strawberry (*Fragaria × ananassa* Duch.), a perennial herb from the Rosaceae family, is valued for its adaptability and widespread cultivation globally. Strawberries are rich in bioactive compounds, including Vitamin c (Vc), phenolics, anthocyanins, galacturonic acid, and strawberry polysaccharides (SPs) [10]. However, fruit quality and nutrient composition are largely influenced by cultivation methods, species, cultivation practices, and geographical conditions [11]. Two primary production systems, conventional soil cultivation and elevated matrix soilless cultivation, have been widely used in strawberry production. Conventional soil planting, regarded as the most traditional approach, relies on loose porous soil that provides good aeration and drainage while supporting a rich microbial ecosystem for nutrient supply [12]. However, the extensive use of chemicals in conventional cultivation can lead to residue accumulation in agroecosystems and crops [13]. In contrast, elevated matrix soilless planting delivers water and nutrients via drip irrigation through a substrate, enabling the precise control of growth conditions and reducing the risk of soil-borne pathogens [14]. This method also conserves land space and allows for more efficient centralized management, making it increasingly popular for strawberry production. Drobek et al. [15] demonstrated that pectin isolated from organic strawberries degrades more rapidly than pectin from conventional strawberries. Although numerous studies have explored the regulation of strawberry growth and fruit quality [16,17], relatively few have focused on the physicochemical properties and biological activities of polysaccharides in strawberry fruits [18]. Moreover, the effects of different planting systems on the physicochemical properties and bioactivities of SPs remain largely unexamined. SPs are a good source of water polysaccharides, but little information is available on their functional and hypoglycemic properties.

Therefore, SPs were extracted in the present study from strawberries cultivated by elevated matrix soilless planting and by conventional soil planting and their chemical composition and structural properties were examined. In addition, the nonenzymatic glycosylation inhibitory ability, glucose-lowering activity, and rheological behavior of SPs were evaluated. These findings will clarify how different strawberry planting systems affect the physicochemical properties of SPs and provide guidance for the potential application of SPs as functional additives or pharmaceutical supplements.

## 2. Materials and Methods

### 2.1. Materials and Chemicals

Fresh strawberry fruits were collected in January 2023 from Jiuan Strawberry Planting Base (Guiyang, China). The strawberry variety selected in this experiment was “Benihoppe”, grown in a greenhouse setting with temperature, humidity, and fertilization managed by automated data systems. Planting was carried out in sunny, fertile soil (or substrate) with good drainage, using seedlings with four leaves. Mounds (40 cm high, 40 cm wide, and 5 m long) were spaced 50 cm apart, and strawberries were planted 15–20 cm apart in both the conventional soil and elevated matrix soilless systems. Soil was used as the planting medium in the conventional system, whereas Coco Coir was used in the elevated matrix soilless system (installed on an H-type elevated frame). For temperature and humidity control, during the bud formation stage, the daytime temperature should be maintained at 25–28 °C, the nighttime temperature at 10–15 °C, and the humidity at around 80%. During the flowering stage, the daytime temperature should be controlled at 23–25 °C, the nighttime temperature at 8–10 °C, and the humidity at around 40%. During the fruiting stage, the daytime temperature should be kept at 20–25 °C, the nighttime temperature at 5–7 °C, and the humidity at around 60%. In addition, an insolation time of 10–12 h is provided during the bud formation period. The sunshine time should be controlled for 12–15 h during the flowering and fruiting periods. The above temperature, humidity, and insolation conditions were applied simultaneously in a conventional soil planting system and elevated matrix soilless planting system. Bovine serum albumin V (BSA, A8020-100g) was obtained from Solarbio Technology Co., Ltd. (Beijing, China). Monosaccharide standards (glucose, galactose, arabinose, rhamnose, mannose, fucose, xylose, galacturonic acid, and glucuronic acid, purity ≥ 99%) were purchased from TOKYO Chemical Industry Co., Ltd. (Tokyo, Japan). Aminoguanidine hydrochloride (AG) was acquired from Macklin Biochemical Co., Ltd. (Shanghai, China). α-Glucosidase (S10050-100U) was purchased from Yeyuan Biotech Co., Ltd. (Shanghai, China), and p-nitrobenzene α-D-pyran glucoside (*p*PNG) was obtained from Aladdin Biochemical Technology Co., Ltd. (Shanghai, China). All other reagents and solvents were of analytical grade and purchased locally.

### 2.2. Preparation of SPs

SPs were extracted from strawberries grown under elevated matrix soilless and conventional soil systems by following a hot water extraction method [19]. Fresh strawberry samples were sliced, hot-air-dried, and powdered. The strawberry powder (1:4, *w*/*v*) was defatted with petroleum ether for 24 h, then treated with 90% ethanol (1:4, *w*/*v*) for another 24 h to obtain pre-treated strawberry powder. Next, 100 g of pre-treated powder was mixed with deionized water at a ratio of 1:20 (*w*/*v*) and extracted at 90 °C for 3 h. The mixture was centrifuged at 4000 rpm for 15 min to obtain the supernatant. The supernatant was then concentrated under reduced pressure, decolorized with Sevag reagent, dialyzed in distilled water using a dialysis membrane (MW cut-off 8000–14,000 Da), further concentrated, and precipitated with 95% ethanol (*v*/*v*) at 4 °C. The precipitates were collected, redissolved in distilled water, and freeze-dried. The resulting polysaccharides were labeled EP-SP (elevated system) and GP-SP (ground system). The yield (%) of SPs was calculated using the following equation:(1)$$\mathrm{SPs}\;\mathrm{yield}\;(\%,w/w)=\frac{\mathrm{weight}\;\mathrm{of}\;\mathrm{dried}\;\mathrm{SPs}\;(g)\;}{\mathrm{weight}\;\mathrm{of}\;\mathrm{pretreated}\;\mathrm{strawberry}\;\mathrm{fruit}\;\mathrm{powder}\;(g)}\times 100$$

### 2.3. Chemical Composition Analysis

The total sugar content in the SPs was determined by the phenol-sulfuric acid method [20], with D-glucose as the standard. Uronic acid content was assessed via the m-hydroxybiphenyl method [21], using D-galacturonic acid monohydrate as the standard. Protein content was measured using the Coomassie brilliant blue method [22], with BSA as the standard.

### 2.4. Monosaccharide Composition and Molecular Weight Analysis

High-performance anion-exchange chromatography (HPAEC) was employed to determine the monosaccharide composition of SPs [23]. SPs (5 mg) were hydrolyzed with 3 mL of trifluoroacetic acid (3 M) at 121 °C for 2 h. Vacuum evaporation followed, along with two methanol washes. The residues were then dissolved in 50 mL of deionized water and filtered through a 0.22 μm membrane. An aliquot of 25 μL was injected into a Dionex ICS-6000 system (Thermo Scientific, Waltham, MA, USA) equipped with an ED electrochemical detector (Au electrode), a Dionex Carbopac PA20 column (3 mm × 150 mm), and a CarboPac PA20 guard column (3 mm × 30 mm). The column temperature was maintained at 30 °C. The mobile phase included deionized water (A), 20 mM NaOH (B), 0.5 M sodium acetate (C), and 0.1 M NaOH (D) at a flow rate of 0.5 mL/min. The gradient elution was as follows: 0–20 min, 10% B and 90% A; 20–40 min, 10% B, 20% C, and 70% A; 40–57 min, 50% D and 50% A. Monosaccharide standards and the SP samples were analyzed under identical conditions.

High-performance gel permeation chromatography (HPGPC) was used to determine molecular weight. An Agilent 1260 chromatograph (Agilent Technologies, Santa Clara, CA, USA) with a TSK gel GMPWXL column (7.8 mm × 300 mm, Tosoh Biotech, Tokyo, Japan) and a refractive index detector (RID) was utilized. Samples (25 μL) were eluted with 0.02 M NaNO_3_ at 0.5 mL/min at 30 °C. Glucan standards (Mw range 9.8–739 kDa) were used to construct a calibration curve, enabling the calculation of weight-average molecular weight (Mw), number-average molecular weight (Mn), and the polydispersity index (Mw/Mn).

### 2.5. Fourier Transform Infrared (FT-IR) Spectroscope

FT-IR spectra were recorded on a Spectrum Two FT-IR spectrometer (PerkinElmer Co., Waltham, MA, USA). The dried SPs were blended with spectrally pure KBr at a 1:100 mg ratio. The absorbance spectra were collected from 4000–500 cm^−1^ with a resolution of 4 cm^−1^.

### 2.6. Congo Red Test

An SP solution (2 mL, 2.5 mg/mL) was mixed with an equal volume of Congo red solution (80 μM). Subsequently, 1.0 M NaOH was added stepwise to adjust NaOH concentration from 0 to 0.50 M (in 0.05 M increments). The mixture was left at 25 °C for 10 min, and the absorption spectrum from 400 to 600 nm was recorded on a Multiskan SkyHigh microplate reader (Thermo Scientific, Waltham, MA, USA). The maximum absorption wavelength was determined.

### 2.7. Differential Scanning Calorimetry (DSC)

The thermal stability of the SPs was assessed using a DSC instrument (DSC-4000, PerkinElmer Co., USA), following published methods [24]. Approximately 6 mg of the sample was sealed in an aluminum crucible, with an empty crucible serving as the reference. The temperature was increased from 45 °C to 445 °C at a rate of 10 °C/min under a nitrogen flow of 20 mL/min.

### 2.8. Particle Size and Potential Determination

The particle size distribution and zeta potential of SP solutions (2 mg/mL) were analyzed by dynamic light scattering (DLS) using an NS-90Z (Malvern Zetasizer Co., Ltd., Worcestershire, UK) at 25 °C.

### 2.9. X-Ray Diffraction (XRD)

The crystalline characteristics of SPs were examined on an X-ray diffractometer (D8 Advance, Bruker, Bremen, Germany) at 40 mA and 40 Kv. Diffraction patterns were recorded over a 2θ range of 5° to 80° with a scanning speed of 5°/min.

### 2.10. Scanning Electron Microscopy (SEM)

Surface microstructures of SPs were visualized using a SIGMA 300 scanning electron microscope (ZEISS, Oberkochen, Germany) at magnifications of 50×, 100×, and 200×. Images were recorded at an acceleration voltage of 5.0 kV.

### 2.11. Non-Enzymatic Glycation Inhibition Assay

#### 2.11.1. BSA–Fructose Reaction Model

All components were dissolved in phosphate buffer (0.2 M, pH 7.4). The reaction mixtures were analyzed subsequently.

The inhibitory effects of SPs on non-enzymatic glycosylation were assessed using a BSA–fructose model following previously described methods [25]. BSA (20 mg/mL) and fructose (500 mM) were mixed with various SP concentrations (1, 2, 3, or 4 mg/mL) or AG (1, 2, 3, or 4 mg/mL), then incubated at 50 °C for 24 h to facilitate glycation. The mixtures were incubated at 50 °C for 24 h to allow glycation. All components were dissolved in phosphate buffer (0.2 M, pH 7.4). The reaction mixtures were analyzed subsequently.

#### 2.11.2. Determination of Fructosamine

Fructosamine was measured based on the reduction of nitroblue tetrazole (NBT) [26]. A mixture containing 40 μL of sample solution and 320 μL of ultrapure water was combined with 1.6 mL of NBT solution (0.3 mM in 100 mM sodium carbonate buffer, pH 10.35) and incubated at 25 °C for 15 min. Absorbance at 530 nm was measured on a UV spectrophotometer (UV-2550, Shimadzu, Tokyo, Japan). Blank and positive controls underwent the same procedure. Fructosamine inhibition (%) was calculated using the following equation:(2)$$Inhibition\;rate\;(\%)=\left(\frac{A_{C}-A_{S}}{A_{C}}\right)\times 100$$
where *A_C_* represents the absorbance of the glycosylation system in the absence of the sample, while *A_S_* is the absorbance after the addition of the sample.

#### 2.11.3. Determination of α-Dicarbonyl Compounds

α-Dicarbonyl compounds were quantified using the Girard-T assay [25]. A mixture of 0.4 mL of reaction solution, 0.2 mL of Girard-T solution (500 mM), and 3.4 mL of sodium formate solution (500 mM, pH 2.9) was incubated at 37 °C for 1 h. The absorbance was read at 290 nm, and α-dicarbonyl compound inhibition (%) was calculated as(3)$$Inhibition\;rate\;(\%)=\left(\frac{A_{C}-A_{S}}{A_{C}}\right)\times 100$$
where *A_C_* is the absorbance of the glycosylation system without the sample, while *A_S_* is the absorbance after the addition of the sample.

#### 2.11.4. Determination of Fluorescent AGEs

Fluorescent AGEs were measured with a F-320 fluorescence spectrophotometer (Gangdong Technology Co., Ltd., Tianjin, China). A mixture of 2 mL phosphate-buffered saline (0.2 M, pH 7.4) and 60 μL of glycation solution was prepared [27]. Fluorescence intensity was recorded at an excitation of 370 nm and emission of 450 nm. Fluorescent AGE inhibition (%) was calculated as(4)$$Inhibition\;rate\;(\%)=\left(\frac{F_{C}-F_{S}}{F_{C}}\right)\times 100$$
where *F_C_* is the fluorescence intensity of the glycosylation system without the sample and *F_S_* is the fluorescence intensity after adding the sample.

### 2.12. α-Glucosidase Inhibition Assay

The inhibition of α-glucosidase by SPs was measured using *p*PNG as the substrate [28]. Phosphate buffer (0.1 M, pH 6.9) was used for the preparation of all samples and reagents. A mixture of 100 μL of SP solution (1, 2, 4, 6, 8 mg/mL) and 100 μL of α-glucosidase (0.5 U/mL) was incubated at 37 °C for 10 min. Next, 100 μL of *p*PNG (5 mM) was added, and the reaction proceeded for 20 min at 37 °C. The reaction was terminated by adding 1 mL sodium carbonate solution (1 M). Absorbance at 405 nm was measured using a multifunctional microplate reader. Acarbose served as the positive control. The inhibition activity (%) of SPs was calculated using the following formula:(5)$$Inhibitory\;activity\;(\%)=\left(1-\frac{A_{S}-A_{C-1}}{A_{C-2}}\right)\times 100$$
where *A_S_* is the reaction mixture containing both enzyme and SPs; *A_C_*_−1_ represents the mixture of SPs and *p*PNG without the enzyme, and *A_C_*_−2_ denotes the control reaction (enzyme and substrate without SPs).

### 2.13. Kinetics of α-Glucosidase Inhibition

The inhibition type was determined by varying substrate (*p*PNG) concentrations (1, 2, 3, 4 mM) and SP concentrations (0, 1, 4, 8 mg/mL) at a constant α-glucosidase level (0.5 U/mL). The reciprocal of initial velocity (1/*V*) was plotted against the reciprocal of substrate concentration (1/[*S*]) in Lineweaver–Burk plots. The Michaelis constant (*K_m_*) and maximum reaction velocity (*V*_max_) were derived from the Michaelis–Menten equation (Equation (6)) [29], and the inhibition constants (*K_is_*) were determined as described by (Equation (7)) [30].(6)$$\frac{1}{V}=\frac{1}{V_{\max}}+\frac{K_{m}}{V_{\max}}\frac{1}{\left[S\right]}$$(7)$$Y-\mathrm{intercept}\;=\frac{1}{V_{\max}}+\frac{K_{m}}{K_{is}V_{\max}}[S]$$

### 2.14. Fluorescence Spectra Analysis

Various concentrations of SPs (0, 1, 2, 4, 6, 8, 10 mg/mL) were mixed with α-glucosidase (0.75 U/mL) at a 1:1 ratio and incubated for 10 min at 37 °C [31]. The fluorescence quenching effect of SPs on α-glucosidase was recorded using a fluorescence spectrophotometer (F-320, Gangdong Technology Co., Ltd., Tianjin, China) from 300 to 500 nm at an excitation wavelength of 280 nm. Excitation and emission slits were set to 5 nm, with a scanning rate 1200 nm/min. The Stern–Volmer plot (Equation (8)) provided the quenching rate constant (*K_q_*) and Stern–Volmer constant (*K_sv_*). The double logarithmic plot (Equation (9)) yielded the binding constant (*K_a_*) and number of binding sites (*N*).(8)$$F_{0}/F=1+K_{q}\tau_{o}[Q]=1+K_{sv}[Q]$$(9)$$log\frac{F_{0}-F}{F}=logK_{a}+N\mathrm{lg}[Q]$$
where *F*_0_ and *F* are the fluorescence intensities without and with SPs, respectively; *τ_0_* is the lifetime of the fluorophore without the quencher (10^−8^ s in this study); and [Q] denotes the concentration of the SPs.

### 2.15. Rheological Measurements

The apparent viscosity of SP solutions (30, 40, 50, 60, 70 mg/mL) was measured using a rheometer (MCR 302e, Anton Paar, Graz, Austria) equipped with parallel steel plates (60 mm diameter, 1.0 mm gap). Flow curves were obtained over a shear rate range of 0.1–1000 s^−1^ at 25 °C. The relationship between the apparent viscosity and shear rate was analyzed using the Cross (Equation (10)) and Carreau–Yasuda (Equation (11)) models [32,33].(10)$$\eta =\eta_{\infty }+(\eta_{0}+\eta_{\infty })/{\left(1+\lambda \gamma \right)}^{m}$$(11)$$\frac{\eta -\eta_{\infty }}{\eta_{0}-\eta_{\infty }}={\left\{1+{\left(\lambda \gamma \right)}^{a}\right\}}^{\frac{\left(n-1\right)}{a}}$$
where *η* is the apparent viscosity (Pa⋅s), *η*_0_ is the zero-shear rate viscosity (Pa⋅s), and *η*_ꝏ_ is the infinite-shear rate viscosity (Pa⋅s). *λ* is a time constant (s) related to the polymer’s relaxation times in solution, defining the onset of the power law region on the shear rate axis. *m* is a dimensionless exponent and *γ* is the shear rate (s^−1^). *a* represents the power governing the transition between low- and high-shear rate regions and *m* is the dimensionless fitting parameter. If the upper Newtonian plateau at higher shear rates (0.01–1000 s^−1^) is not reached, *η*_ꝏ_ can be neglected.

Based on the apparent viscosity test results, the linear viscoelastic regime (LVR) for each sample was determined using a strain sweep (0.1–120%) at a constant oscillation frequency of 1 Hz. The LVR for each sample was established, and the 1% strain was selected. The storage modulus (*G*′) and loss modulus (*G*″) of EP-SP and GP-SP were then measured using an oscillation frequency sweep from 1 to 100 rad/s at 25 °C. The slope values of *G*′ and *G*″ during the frequency sweep were calculated using the power law model (Equations (12) and (13)).(12)$$G^{\prime }=k^{\prime }\omega^{n^{\prime }}$$(13)$$G^{\prime \prime }=k^{\prime \prime }\omega^{n^{\prime \prime }}$$
where *ω* is the angular frequency, *n*′ (*n*″) is the frequency exponent, and *k*′ (*k*″) denotes the intercepts.

### 2.16. Statistical Analysis

All experiments were performed in triplicate, and results are presented as mean ± standard deviation (SD). A one-way ANOVA (*p* < 0.05) was conducted using SPSS 20 (SPSS Inc., Chicago, IL, USA), and Tukey’s test was used to compare mean values. Significant differences (*p* < 0.05) among treatments are indicated by different superscripts. Graphs were produced in Origin 2019 (OriginLab Corporation, Northampton, MA, USA).

## 3. Results and Discussion

### 3.1. Extraction Yields and Chemical Composition Analysis

The extraction yield and chemical composition (including total sugar, uronic acid, and protein) of SPs from different planting systems are summarized in Table 1. The yield of EP-SP (5.88%) was higher than that of GP-SP (4.67%), indicating that planting systems influence SP yield. Similar effects have been observed in studies on Ganoderma sinense [34] and Cochlospermum tinctorium [35]. The uronic acid content of EP-SP and GP-SP was 22.06% and 22.28%, respectively, showing no significant effect of planting systems on uronic acid content. However, the protein and total sugar contents differed, with EP-SP (1.65% and 39.38%) exhibiting lower values than GP-SP (2.12% and 34.92%). These results confirm that different planting systems can affect the yield and chemical composition of SPs.

Figure 1A presents the molecular weight chromatograms of SPs. The weight-average molecular weight (Mw) of SPs was calculated using a molecular weight standard curve, yielding values of 632.10 kDa for EP-SP and 611.88 kDa for GP-SP, classifying both as high-molecular-weight polysaccharides. Compared with other fruit polysaccharides, such as orange (68.00 kDa) [36], watermelon (30.20 kDa) [37], banana (1470–2506 kDa) [38], and lychee (4.06–85.60 kDa) [39], both SP samples fall within the medium–high-molecular-weight range.

Differences in molecular mass distribution were assessed through the polydispersity index (Mw/Mn) (Table 1), with higher values indicating a broader molar mass distribution [24]. Both EP-SP and GP-SP exhibited identical polydispersity indexes (3.93), indicating a wide molecular mass distribution. These results suggest that planting systems influence the molecular weight of SPs, potentially impacting biological activity. Further investigation is needed to determine whether the bioactivity of EP-SP and GP-SP differs.

### 3.2. Monosaccharide Composition of SPs

Monosaccharides, the basic structural units of polysaccharides, determine their biological activities [40]. The monosaccharide composition of EP-SP and GP-SP (Figure 1B and Table 1) included fucose (Fuc), rhamnose (Rha), arabinose (Ara), galactose (Gal), glucose (Glc), xylose (Xyl), mannose (Man), galacturonic acid (GalA), and gluconic acid (GlcA). It was shown that planting systems did not alter monosaccharide types but did influence their molar ratios. SPs were primarily composed of Rha, Gal, Glc, and GalA. GP-SP contained higher Gal (11.06%) and total uronic acid (19.66%, with GalA at 19.12%), while EP-SP had higher Rha (12.33%) and Glc (49.29%). Minimal differences were observed in total uronic acid between the two samples, aligning with the chemical composition results. It was thus concluded that planting systems influence monosaccharide ratios without altering the overall monosaccharide composition.

### 3.3. Characterizations and Morphology of SPs

#### 3.3.1. Congo Red Test Analysis

The Congo red test was employed to assess the triple-helix conformation of both SPs. Congo red forms a complex with polysaccharides that have a triple-helix structure, causing a red shift in the maximum absorption wavelength (λ_max_) under weakly alkaline conditions [41]. As shown in Figure 1C, both EP-SP and GP-SP exhibited a λ_max_ at a NaOH concentration of 0.05 mol/L, reflecting a triple-helix conformation. With higher NaOH concentrations, λ_max_ gradually declined, presumably due to the disruption of hydrogen bonds, which breaks the triple-helix structure and prevents complex formation with Congo red [42].

#### 3.3.2. FT-IR Spectroscope Analysis

FT-IR spectroscopy was used to identify the structural characteristics of SPs (Figure 1D). Both EP-SP and GP-SP samples exhibited characteristic polysaccharide peaks at 3414, 2923, 1746, 1051, and 831 cm^−1^. The broad peak near 3414 cm^−1^ was attributed to O-H stretching vibrations, and the peak at 2923 cm^−1^ was assigned to C-H stretching. The peak at 1746 cm^−1^ indicated C=O stretching in esterified carboxyl groups (-COOR), suggesting the presence of uronic acid, in line with prior chemical analyses. The region of 800–1200 cm^−1^ represented the carbohydrate fingerprint region, with the 1051 cm^−1^ indicating C-O stretching in pyranose rings. Additionally, the peak at 831 cm^−1^ implied a pyranose form of the α-type glycosidic bond [43]. The results indicate that planting systems did not affect the primary functional groups of SPs, consistent with previous findings for wolfberry polysaccharides [44].

#### 3.3.3. Thermal Characteristics

The DSC curve recorded the heat flow change in the SPs during the heating process. As shown in Figure 2A and Table 2, both SPs exhibited an endothermic and an exothermic peak. Thermal stability parameters, including the melting temperature (*Tm*), melting enthalpy (Δ*Hm*), degradation temperature (*Tg*), and degradation enthalpy (Δ*Hg*), were obtained. EP-SP and GP-SP presented endothermic peaks at 159.87 °C and 139.08 °C, respectively, likely due to dehydration, peripheral polysaccharide chain loss, or dehydroxylation reactions [45]. The magnitude of Δ*Hg* reflects the molecular arrangement order, while a higher *Tg* indicates greater thermal stability [24,46]. The degradation peaks appeared at 226.17 °C for EP-SP and 225.70 °C for GP-SP, caused by thermal decomposition or the depolymerization of polysaccharides [47]. The small difference in *Tg* (0.47 °C) indicates similar thermal stability, whereas the Δ*Hg* values (13.93 J/g for EP-SP and 18.74 J/g for GP-SP) indicated a more ordered molecular arrangement in GP-SP. Overall, DSC analysis showed that while planting systems did not affect thermal stability, they did influence molecular arrangement.

#### 3.3.4. Particle Size and Zeta Potential Analysis

Particle size and zeta potential are indicators of solution stability [48]. As shown in Figure 2B, EP-SP had a particle size of 1121.00 nm, while GP-SP measured 1073.33 nm. Polysaccharides with larger molecular weights tend to form larger particles in aqueous solutions [49], consistent with the measured molecular weights. Due to the presence of uronic acid, EP-SP and GP-SP exhibited a negative charge in solution [50], with zeta potentials of −24.23 mV for EP-SP and −25.07 mV for GP-SP. These results suggest similar particle sizes and negative charges for both SP samples, indicating comparable stability in aqueous solution regardless of planting systems.

#### 3.3.5. XRD and SEM Analysis

XRD has frequently been used to determine the crystalline or amorphous properties of SPs. As shown in Figure 2C, diffraction peaks appeared at 23.03° and 37.06° for EP-SP and 21.09° and 35.85° for GP-SP within the 5–80° 2θ range. The decreased diffraction intensity of EP-SP suggested that the elevated matrix soilless planting system influences polysaccharide chain arrangement, thus enhancing crystallinity [39]. Sea buckthorn polysaccharides have also been reported to exhibit semi-crystalline properties, although with slightly different peak positions, likely reflecting variations in raw materials [49]. Because polysaccharide chain arrangements affect solubility, swelling, and flexibility, it was confirmed that planting systems can alter certain physical properties of SPs [51].

SEM images revealed aggregated sheet-like structures with smooth surfaces in both EP-SP and GP-SP at magnifications of 50×, 100×, and 200× (Figure 3). The relatively smooth morphology may be attributed to higher neutral sugar content [49]. These findings confirm that planting systems did not significantly alter the morphological characteristics of SPs.

### 3.4. Non-Enzymatic Glycation Inhibition Activity

The inhibitory effects of SPs on non-enzymatic glycation were evaluated by examining the formation of characteristic products in three stages. In the early stage, the formation of fructosamine was assessed in BSA–fructose mixtures incubated with SPs or AG (Figure 4A). The IC_50_ values for EP-SP, GP-SP, and AG were 7.82 mg/mL, 7.47 mg/mL, and 2.81 mg/mL, respectively, indicating slightly better fructosamine inhibition by GP-SP than EP-SP, although both were weaker than AG.

During the intermediate stage, α-dicarbonyl compounds are formed through the oxidation and dehydration of Amadori products, leading to stable AGEs via protein crosslinking [42]. The inhibition of these compounds is crucial, as they are highly reactive and readily form AGEs [52]. As shown in Figure 4B, EP-SP and GP-SP achieved maximum inhibition rates of 75.32% and 72.32%, respectively, lower than AG (89.02%). The IC_50_ values were 5.86 mg/mL for EP-SP and 6.40 mg/mL for GP-SP, compared to 2.20 mg/mL for AG. Notably, EP-SP demonstrated a better inhibitory effect on α-dicarbonyl compounds than GP-SP, possibly due to its higher Mw. This finding aligns with previous research on blackberry fruit polysaccharides, where a higher Mw correlated with an enhanced inhibition of α-dicarbonyl compound formation [25].

In the final stage, AGEs can be classified as fluorescent or non-fluorescent, with their content assessed via fluorescence intensity [53]. Figure 4C shows that both SPs inhibited fluorescent AGE formation in a concentration-dependent manner, with IC_50_ values of 11.87 mg/mL (EP-SP) and 11.18 mg/mL (GP-SP) compared to AG (0.67 mg/mL). GP-SP exhibited a slightly stronger inhibition of fluorescent AGEs. SPs provided the most pronounced inhibition in the intermediate stage, suggesting that their main anti-glycation effect occurs during α-dicarbonyl compound formation. Although GP-SP generally demonstrated stronger anti-glycation capacity, EP-SP exceeded GP-SP in α-dicarbonyl compound inhibition, possibly because the GP-SP concentration was inadequate to fully block carbonyl groups on Amadori products. These data indicate that both SPs possess significant anti-glycation activities, likely due to their molecular weight, monosaccharide composition, and glycosidic linkage characteristics [42,54]. The results overall suggest that GP-SP exhibits robust anti-glycation effects, implying that ground planting can enhance the anti-glycation abilities of polysaccharides.

As shown in Figure 4D–F, the BSA–glucose mixture exhibited strong fluorescence intensity, confirming the presence of fluorescent AGEs. A dose-dependent decline in fluorescence intensity was observed upon the addition of SPs and AG, accompanied by a red shift in the maximum emission wavelength from 441.4 nm to 444.2 nm for EP-SP, from 442.6 nm to 444.2 nm for GP-SP, and from 447 nm to 450.2 nm for AG. This behavior indicates increased polarity around the fluorophore, consistent with previous findings on galangin polysaccharides in BSA–glucose models [26].

### 3.5. Hypoglycemic Activities

#### 3.5.1. Analysis of Inhibition Rate and Inhibition Kinetics

α-glucosidase is a key hydrolase in the small intestine, and inhibiting its activity can reduce postprandial blood glucose levels. The study of α-glucosidase inhibitors is thus considered an effective strategy for diabetes mellitus treatment. As shown in Figure 5A, both EP-SP and GP-SP inhibited α-glucosidase in a dose-dependent manner (1–10 mg/mL). The IC_50_ values of EP-SP (2.7377 mg/mL) and GP-SP (2.4583 mg/mL) were higher than that of acarbose (0.07 mg/mL), indicating that both SPs were less potent than acarbose. However, plant-derived polysaccharides such as SPs exhibit negligible side effects and possess antioxidant properties, suggesting potential for anti-diabetic drug development. It is likely that the stronger inhibitory activity of GP-SP is related to its elevated uronic acid content and lower molecular weight, which may facilitate binding to the α-glucosidase active site [55].

To elucidate the potential of SPs as α-glucosidase inhibitors, the kinetics and mechanism were investigated. The double-reciprocal Lineweaver–Burk plots and relevant kinetic parameters are shown in Figure 5B,C and Table 3, respectively. It was found that α-glucosidase without inhibitors had *K_m_* and *V_max_* values of 5.0845 mM and 0.1495 ΔA405 min^−1^ for EP-SP and 5.5557 mM and 0.1635 ΔA405 min^−1^ for GP-SP. Upon the addition of 1, 5, and 10 mg/mL of EP-SP, the *K_m_* values decreased to 2.7083, 1.7716, and 0.8615 mM, and the *V_max_* values decreased to 0.0800, 0.0531, and 0.0238 ΔA405 min^−1^, respectively. Similarly, with 1, 5, and 10 mg/mL of GP-SP, the *K_m_* values decreased to 2.7215, 2.1084, and 0.8383 mM, and the *V_max_* values decreased to 0.0789, 0.0605, and 0.0229 ΔA405 min^−1^, respectively. Both *K_m_* and *V_max_* decreased with increasing inhibitor concentration, while the slope of the inhibitory kinetic curves (*K_m_*/*V_max_*) remained nearly constant. These results indicate that SPs act as uncompetitive inhibitors of α-glucosidase, binding only to the enzyme–substrate complex and not to the free enzyme. Specifically, the formation of an enzyme–substrate–inhibitor ternary complex prevents substrate turnover, thereby inhibiting enzyme activity [56]. Natural inhibitors of α-glucosidase exhibit competitive, non-competitive, or mixed inhibition types [57]. Uncompetitive inhibitors are considered advantageous for drug development compared to competitive or non-competitive inhibitors due to potentially enhanced in vivo efficacy [58].

The inhibition constants (*K_is_*) were obtained from the Y-intercepts of the Lineweaver–Burk plots. A smaller *K_is_* value indicates a stronger inhibitory effect, since the enzyme–substrate complex is more readily bound [59]. EP-SP and GP-SP exhibited *K_is_* values of 2.0027 mg/mL and 1.6682 mg/mL, respectively, indicating a stronger affinity of GP-SP for the α-glucosidase enzyme–substrate complex, in agreement with its higher inhibition rate. The kinetic results showed that while the different planting systems did not affect the inhibition mechanism of α-glucosidase by SPs, they did influence the inhibition potency.

#### 3.5.2. Fluorescence Quenching Analysis

Fluorescence quenching occurs when an inhibitor interacts with an enzyme, causing changes in protein polarity and leading to the unfolding of fluorescent amino acid residues, which results in decreased fluorescence intensity [4]. In this study, changes in α-glucosidase fluorescence intensity were measured in the absence and presence of varying SP concentrations (1–10 mg/mL). As shown in Figure 5D,E, the fluorescence intensity decreased with increasing amounts of polysaccharide (1–10 mg/mL), confirming that SPs can quench the fluorescence of α-glucosidase and validating the interaction between them. Furthermore, the fluorescence intensity curve did not exhibit a significant red shift with increasing SP concentration, indicating that SPs do not alter the internal microenvironment or spatial conformation of α-glucosidase, consistent with reports on other α-glucosidase inhibitors [60,61].

Mechanistically, fluorescence quenching can be dynamic, static, or a combination of both. The quenching mechanism can be determined by analyzing the magnitude of the quenching constant (*K_q_*) [62]. The *K_q_* values for EP-SP and GP-SP, obtained from Table 3, were 4.1204 × 10^12^ M^−1^·s^−1^ and 6.8077 × 10^12^ M^−1^·s^−1^, respectively. Since these values exceed the maximum dynamic fluorescence quenching rate constant (2.0 × 10^10^ M^−1^·s^−1^), the binding mechanism is indicated to be static quenching [63]. The binding constants (*K_a_*) and number of binding sites *(N*) were calculated (Figure 5F,G), revealing that GP-SP (*Ka* = 5.1352 × 10^5^ M^−1^) had a stronger binding affinity to α-glucosidase than EP-SP (*Ka* = 3.3124 × 10^4^ M^−1^). The *N* values for EP-SP (0.9830) and GP-SP (1.1945) were approximately one, indicating a single binding site on α-glucosidase [64]. These results agree with the observed inhibition rates and kinetic data, demonstrating that GP-SP exhibits more potent α-glucosidase inhibition. This indicates that different planting systems can influence the hypoglycemic activity of SPs.

Previous studies have shown that polysaccharides can alleviate diabetes through various hypoglycemic mechanisms, including delaying carbohydrate digestion and glucose absorption, inhibiting digestive enzyme activity, exhibiting antioxidant activity, regulating lipid metabolism, modulating AGE levels, and influencing intestinal flora [65]. In this study, two aspects were targeted, which were inhibiting digestive enzyme activity and regulating AGE levels. SPs demonstrated a strong ability to inhibit α-glucosidase, delaying the absorption and digestion of carbohydrates in the digestive tract, thereby reducing postprandial blood glucose levels and controlling hyperglycemia to mitigate complications [1]. Additionally, in vitro anti-glycosylation experiments showed that SPs have significant potential for the anti-glycation and inhibition of AGE production. Elevated AGE levels may signal diabetes and increase the risk of microvascular complications [65,66]. Based on these findings, SPs appear to be promising candidates for diabetes treatment, offering theoretical support for disease prevention and management.

### 3.6. Rheological Properties

#### 3.6.1. Steady Shear Flow Properties

Figure 6A,B illustrates the steady-state flow curves of SP solutions. The apparent viscosity (*η*) increased with rising SP concentrations over the shear rate (*γ*) range of 0.1–1000 s^−1^, indicating the formation of a denser network structure that elevated the shear stress required to flow [67]. When *γ* increased, *η* decreased gradually, likely because SP molecules aligned along the flow direction, reducing intermolecular entanglements. As *γ* increased, the *η* of SP solutions at different concentrations decreased gradually [68]. For instance, under a shear rate of 0.1 s^−1^, the *η* values for 30 mg/mL aqueous solutions of EP-SP and GP-SP were 117.53 mPa·s and 95.31 mPa·s, respectively. When the mass concentration increased to 70 mg/mL, the *η* values rose significantly to 3477.30 mPa·s for EP-SP and 2436.20 mPa·s for GP-SP. The higher *η* of EP-SP compared to GP-SP at the same *γ* and concentration may be due to its higher total sugar content, which enables tighter cross-linking structures and its higher molecular weight [69].

Within the shear rate range of 0.1–1000 s^−1^, both SP samples exhibited two distinct phases, a constant viscosity at lower *γ* (Newtonian plateau) and a decreasing viscosity at higher *γ* (shear-thinning region). The constant apparent viscosity in the low-*γ* region suggests that the rate of intermolecular disentanglements induced by shear forces is nearly balanced with the rate of new entanglements forming. In contrast, in the shear-thinning region, the rate of disentanglements surpasses the rate of new entanglement formation, leading to a decrease in *η* [70]. As depicted in Figure 6, the rapid decline in *η* with increasing *γ* indicates that both SP samples are pseudoplastic fluids exhibiting shear-thinning behavior. Additionally, the shear-thinning zone progressively shifts to lower-*γ* regions as the sample concentration increases, indicating an expanding shear-thinning interval for SPs. Similar trends have been observed in other polysaccharide studies, such as those involving *Sophora alopecuroides* L. seeds [71] and Adansonia digitata leaves [72].

To better understand the shear-thinning behavior of SPs, the *η*-*γ* curves were fitted using the Cross and Carreau–Yasuda models (Table 4), revealing excellent fit quality (R^2^ > 0.98). The zero-rate viscosity (*η*_0_) and time constants (*λ*), which are related to the relaxation time of the biopolymer structure, were examined in both models. With an increase in polysaccharide mass concentration from 20 mg/mL to 70 mg/mL, *η_0_* and *λ* of both EP-SP and GP-SP increased progressively (e.g., *η_0_* of EP-SP increased from 44.43 mPa·s to 3419.22 mPa·s, and *λ* increased from 0.03 × 10^−2^ to 1.75 × 10^−2^ in the Cross model). The relaxation time indicates the rate of viscosity reduction under shear stress; higher *λ* values signify a greater relaxation time and a faster disruption of the structure [73]. Combined with Table 4, it is inferred that the structural damage of EP-SP when rheological behavior occurs in aqueous solution is higher than that of GP-SP, which is more obvious after increasing the concentration.

#### 3.6.2. Dynamic Oscillatory Shear Properties

It is necessary to identify the linear viscoelastic region (LVR) before measuring oscillatory rheological properties to prevent sample disruption [72]. As shown in Figure 6C, the storage modulus (*G*′) and loss modulus (*G*″) remained constant within the LVR but decreased upon exceeding the critical strain (*γ_c_*), reflecting permanent alterations in the intermolecular network [71]. Therefore, a 1% strain was selected to study the dynamic rheological properties of the two SPs.

The viscoelastic behavior of EP-SP and GP-SP was characterized by measuring the frequency dependence of *G*′, *G*″, and tan δ at the same concentration (70 mg/mL) and temperature (25 °C) (Figure 6D,E). Generally, when *G*′ > *G*″, materials exhibit elastic properties characteristic of gels, whereas when *G*′ < *G*″, they display viscous properties and behave more “liquid-like”. Gels can be classified as weak or strong. A dominant *G*′ over *G*″ indicates strong gel properties, consisting of stable intermolecular associations. Conversely, when *G*′ is close to and just above *G*″, the material shows typical weak gel properties, suggesting weak junction zones that can be readily disrupted at low shear rates [74].

Over the measured frequency range (1–100 rad s^−1^), both *G*′ and *G*″ of the SP solutions increased with angular frequency (ω). For both SP samples, *G*′ was consistently higher than *G*″ across the entire ω range, with no crossover observed, indicating strong gel behavior. The loss tangent (tan δ), defined as *G*″/*G*′, provides an intuitive reflection of the dynamic viscoelastic behavior of fluids. A tan δ > 1 indicates predominant viscous characteristics, while tan δ < 1 suggests predominant elastic characteristics [75,76]. As shown in Figure 6F, tan δ for both EP-SP and GP-SP fluctuated with frequency but remained less than one across the entire angular frequency range, indicating that both SP samples exhibit elastic characteristics and can be regarded as elastic gels.

Power law modeling was used to analyze the frequency dependence of *G*′ and *G*″ (Table 5). The intercepts (*k*′, *k*″) and frequency indices (*n*′, *n*″) are commonly used to describe the viscoelastic characteristics of food materials. Generally, a small n value characterizes an elastic gel, while an n value close to one indicates a viscous gel. When *n*′ and *n*″ are close to zero, the system shows low-frequency dependence, and *n*″ usually follows a trend similar to *n*′ [77]. The correlation coefficients (R^2^) between *G*′ or *G*″ and ω for both EP-SP and GP-SP were higher than 0.98 (Table 5), indicating good fits to the power law model. In this experiment, the *n*′ and *n*″ values for EP-SP were 0.6792 and 0.9487, respectively, while for GP-SP, they were 0.6416 and 0.7845. The relatively small *n*′ values, associated with the strength and structure of the gel, indicate that both EP-SP and GP-SP are physical gels characterized as elastic and highly dependent on frequency. Furthermore, the higher *n*′ and *n*″ values of EP-SP compared to GP-SP suggest that EP-SP samples have a higher frequency dependence. This may be attributed to the increased molecular entanglement of EP-SP due to its higher Mw, leading to the formation of elastic active regions and enhanced viscoelastic behavior [50]. In both samples, *k*′ exceeded *k*″, further confirming gel-like behavior.

## 4. Conclusions

In this study, SPs were extracted from strawberries grown under elevated matrix soilless and conventional soil methods, and their physicochemical properties and bioactivities were investigated. The two SPs differed in yield, total sugar content, molecular weight, monosaccharide composition, anti-glycosylation activity, and glucose-lowering activity. Both SPs possessed similar functional groups, triple-helix conformations, semi-crystalline structures, and elastic gel-like behaviors. However, differences were observed in molecular organization and viscoelastic behavior. EP-SP exhibited a higher molecular weight and total sugar content, leading to greater viscosity and more pronounced frequency dependence in rheological properties. The thermal stability of both SPs was comparable, but GP-SP demonstrated a more ordered molecular arrangement. Bioactivity assays indicated that both SP samples exhibited significant non-enzymatic glycation inhibition and hypoglycemic activities. GP-SP showed stronger inhibition of α-glucosidase and a greater ability to inhibit the formation of AGEs, suggesting that conventional soil planting systems may enhance the anti-glycation and hypoglycemic effects of SPs. Fluorescence quenching studies confirmed that both SP samples interact with α-glucosidase through static quenching mechanisms, with GP-SP exhibiting a stronger binding affinity. Overall, the findings suggest that planting systems significantly affect the physicochemical properties and bioactivities of SPs. The superior hypoglycemic and anti-glycation activities of GP-SP underscore its potential as a functional additive or pharmaceutical supplement for diabetes management. These results provide valuable insights for the application of SPs in the food and pharmaceutical industries and underscore the importance of selecting appropriate planting practices to optimize the functional properties of plant polysaccharides.

## Figures and Tables

**Figure 1 foods-14-00238-f001:**
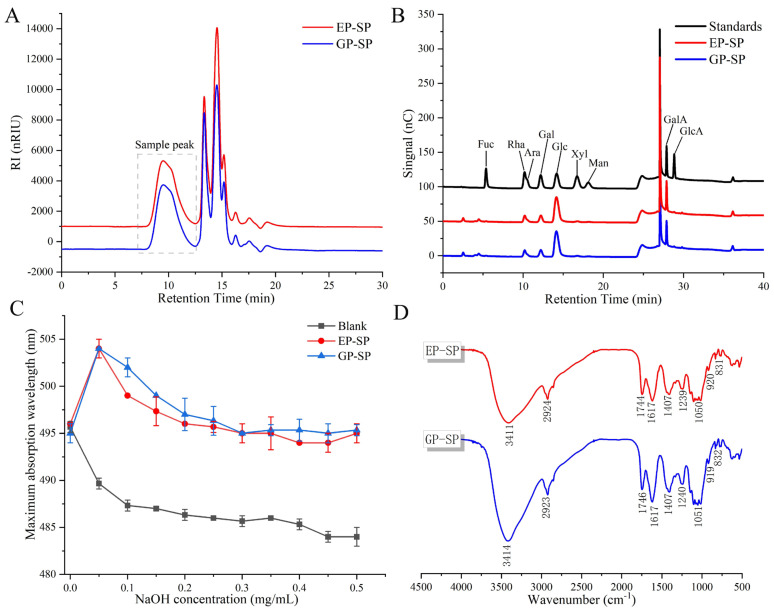
(**A**) HPGPC chromatograms of SPs; (**B**) HPAEC chromatograms of SPs; (**C**) maximum absorption wavelength (λmax) of Congo red–polysaccharide complexes at varying NaOH concentrations for SPS; (**D**) FT-IR spectra of SPs.

**Figure 2 foods-14-00238-f002:**
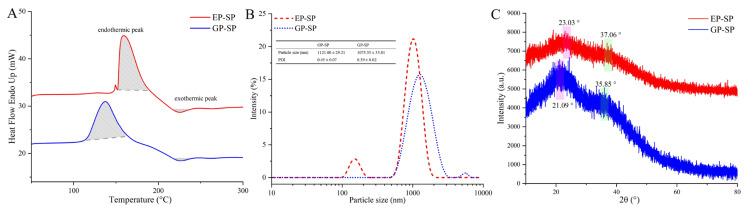
(**A**) DSC curves of SPs; (**B**) particle size of SPs; (**C**) XRD spectra of SPs.

**Figure 3 foods-14-00238-f003:**
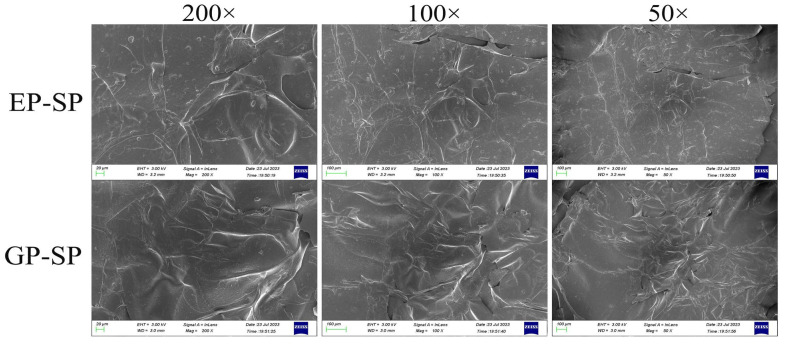
SEM images of EP-SP and GP-SP at magnifications of 200×, 100×, and 50×.

**Figure 4 foods-14-00238-f004:**
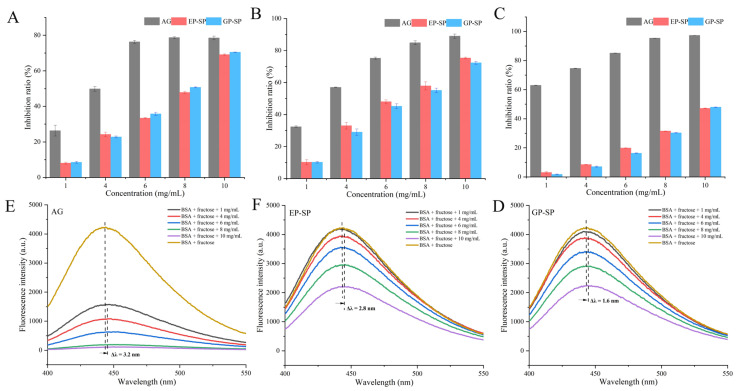
(**A**) Fructosamine inhibition of SPs; (**B**) α-dicarbonyl compound inhibition of SPs; (**C**) AGE inhibition of SPs; (**D**–**F**) fluorescence emission spectra of BSA in the BSA–fructose reaction model in the presence of different concentrations of AG, EP-SP, and GP-SP.

**Figure 5 foods-14-00238-f005:**
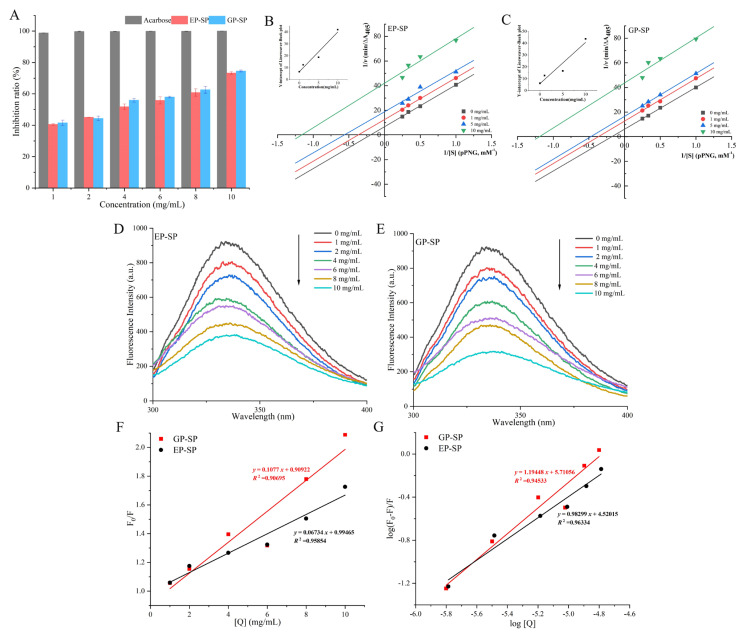
α-glucosidase inhibitory activity of EP-SP and GP-SP (**A**); Lineweaver–Burk plots of α-glucosidase inhibition by SPs and the linear relationship between the Y-intercepts of the Lineweaver–Burk plots and SP concentration (**B**,**C**); fluorescence emission spectra α-glucosidase in the presence of various concentrations of SPs (**D**,**E**); Stern–Volmer plots of α-glucosidase fluorescence quenching by SPs (**F**); plots of log [(F_0_ − F)/F] versus log [Q] for the interaction of SPs and α-glucosidase (**G**).

**Figure 6 foods-14-00238-f006:**
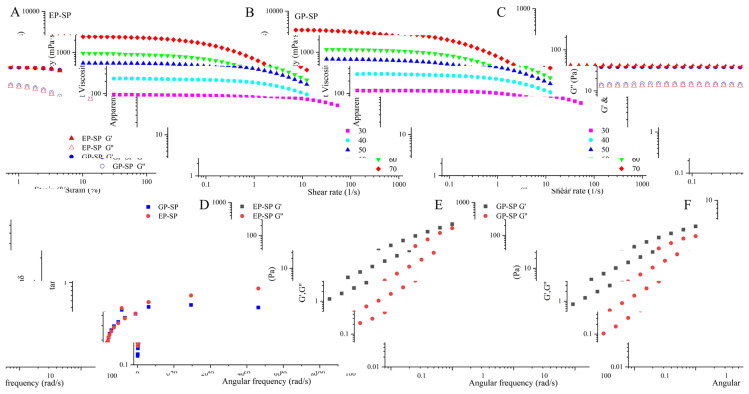
Rheological properties of EP-SP and GP-SP. (**A**,**B**) Apparent viscosity; (**C**) storage modulus (*G*′) and loss modulus (*G*′′) as functions of strain; (**D**,**E**) storage modulus (*G*′) and loss modulus (*G*″) against angular frequency; (**F**) variation in loss tangent (tan δ) with angular frequency.

**Table 1 foods-14-00238-t001:** Extraction yield, chemical composition, sugar composition, and molecular weight distribution of different planting systems for strawberry.

	EP-SP	GP-SP
Yield (%)	5.88 ± 0.87 ^a^	4.67 ± 1.33 ^b^
Protein	1.65 ± 0.34 ^b^	2.12 ± 0.42 ^a^
Uronic acid	22.33 ± 1.47 ^a^	22.44 ± 2.03 ^a^
Total sugar	39.38 ± 0.59 ^a^	34.92 ± 1.43 ^b^
Fucose	0.19 ± 0.11 ^a^	0.35 ± 0.49 ^a^
Rhamnose	12.33 ± 1.54 ^a^	11.83 ± 1.49 ^a^
Arabinose	5.90 ± 0.69 ^a^	4.99 ± 0.65 ^a^
Galactose	10.91 ± 1.81 ^a^	11.06 ± 1.81 ^a^
Glucose	49.29 ± 1.31 ^a^	47.90 ± 2.39 ^b^
Xylose	1.69 ± 0.31 ^a^	2.00 ± 0.20 ^a^
Mannose	2.09 ± 0.46 ^a^	2.20 ± 0.22 ^a^
Galacturonic Acid	17.00 ± 2.78 ^b^	19.12 ± 3.08 ^a^
Gluconic Acid	0.60 ± 0.14 ^a^	0.54 ± 0.09 ^a^
Mw (kDa)	632.10 ± 3.61 ^a^	611.88 ± 6.93 ^b^
Mn (kDa)	161.11 ± 8.82 ^a^	156.10 ± 7.83 ^b^
Mw/Mn	3.93 ± 0.24 ^a^	3.93 ± 0.16 ^a^

Values with different letters on the same row indicate significant differences (*p* < 0.05).

**Table 2 foods-14-00238-t002:** DSC curve parameters for strawberries obtained by different planting systems.

	EP-SP	GP-SP
*Tm* (°C)	159.87 ± 0.57 ^a^	139.08 ± 15.42 ^b^
Δ*Hm* (J/g)	231.02 ± 22.73 ^a^	187.36 ± 40.99 ^b^
*To* (°C)	214.99 ± 0.91 ^a^	210.33 ± 1.36 ^a^
*Tg* (°C)	226.17 ± 0.60 ^a^	225.70 ± 2.19 ^a^
Δ*Hg* (J/g)	13.93 ± 2.18 ^b^	18.74 ± 2.24 ^a^

Values with different letters on the same row indicate significant differences (*p* < 0.05).

**Table 3 foods-14-00238-t003:** Kinetic parameters of α-glucosidase inhibition in the presence of SPs.

Sample	EP-SP				GP-SP			
IC_50_	2.7377 ± 0.12				2.4583 ± 0.19			
Concentration (mg/mL)	0	1	5	10	0	1	5	10
*K_m_* (mM)	5.0845	2.7083	1.7716	0.8615	5.5557	2.7215	2.1084	0.8383
*V_max_* (ΔA405 min^−1^)	0.1495	0.0800	0.0531	0.0238	0.1635	0.0789	0.0605	0.0229
Inhibition type	uncompetitive inhibition				uncompetitive inhibition			
*K_is_* (mg/mL)	2.0027				1.6682			
*K_q_* (M^−1^ s^−1^)	4.1204 × 10^12^				6.8077 × 10^12^			
*K_SV_* (M^−1^)	4.1204 × 10^4^				6.8077 × 10^4^			
*K_α_* (M^−1^)	3.3124 × 10^4^				5.1352 × 10^5^			
*N*	0.9830				1.1945			

**Table 4 foods-14-00238-t004:** Rheological parameters of Cross and Carreau–Yasuda models for SP solutions at different concentrations (20–70 mg/mL).

Models	Concentration (mg/mL)	Cross				Carreau–Yasuda				
		η_0_	λ	m	R^2^	η_0_	λ	a	n	R^2^
EP-SP	20	44.43	0.03 × 10^−2^	0.49	0.9824	44.45	0.02 × 10^−2^	0.48	0.41	0.9830
	30	122.16	0.10 × 10^−2^	0.54	0.9890	117.28	0.14 × 10^−2^	0.76	0.36	0.9996
	40	299.04	0.24 × 10^−2^	0.61	0.9970	296.06	0.20 × 10^−2^	0.69	0.28	0.9995
	50	681.19	0.47 × 10^−2^	0.61	0.9982	707.97	0.25 × 10^−2^	0.56	0.24	0.9984
	60	1263.91	0.80 × 10^−2^	0.62	0.9984	1268.05	0.52 × 10^−2^	0.59	0.27	0.9993
	70	3419.22	1.75 × 10^−2^	0.63	0.9991	3715.92	1.17 × 10^−2^	0.56	0.27	0.9997
GP-SP	20	48.46	0.04 × 10^−2^	0.69	0.9855	48.47	0.03 × 10^−2^	0.68	0.26	0.9856
	30	93.26	0.07 × 10^−2^	0.68	0.9953	92.95	0.06 × 10^−2^	0.70	0.21	0.9968
	40	230.67	0.16 × 10^−2^	0.67	0.9967	231.50	0.06 × 10^−2^	0.61	0.22	0.9974
	50	554.00	0.33 × 10^−2^	0.74	0.9999	543.52	0.22 × 10^−2^	0.63	0.24	0.9989
	60	949.50	0.62 × 10^−2^	0.62	0.9985	954.97	0.32 × 10^−2^	0.58	0.23	0.9990
	70	2425.54	1.06 × 10^−2^	0.66	0.9994	2369.90	1.03 × 10^−2^	0.65	0.33	0.9993

**Table 5 foods-14-00238-t005:** The power law model was used to fit the obtained data (*G*′ and *G*″ vs. ω).

		EP-SP	GP-SP
*G*′ = *k*′ω^n′^	*k*′	9.1322	10.3313
	*n*′	0.6792	0.6416
	R^2^	0.9973	0.9960
*G*″ = *k*″ω^n″^	*k*″	1.7953	3.3329
	*n*″	0.9487	0.7845
	R^2^	0.9954	0.9824

## Data Availability

The original contributions presented in the study are included in the article; further inquiries can be directed to the corresponding author.

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
