# Peer review of "Effect of Planting Systems on the Physicochemical Properties and Bioactivities of Strawberry Polysaccharides"

_foods, 2025, doi:10.3390/foods14020238_

Round 1

Reviewer 1 Report

Comments and Suggestions for Authors

The work in my opinion is very good. I have a few comments on the issue of presentation of the results.  Figure 1 A and B retention time is once written as “retention time” and in the other case abbreviated RT

In Figure 2 the zeta potential is expressed without standard deviation. I would not make the zeta potential data (two values) as much as a graph.

Table 4 a number with two decimal places to the eta would suffice.

Author Response

We greatly appreciate your constructive suggestions and comments, which have significantly enhanced our manuscript. We are resubmitting our revised manuscript titled "Effect of Planting Systems on the Physicochemical Properties and Bioactivities of Strawberry Polysaccharides". We have meticulously revised our manuscript based on the feedback from the editor and reviewers, clarifying and expanding explanations as needed. Changes have been highlighted in red in the revised manuscript. Below are our responses to the comments:

Specific comments

The work in my opinion is very good. I have a few comments on the issue of presentation of the results. Figure 1 A and B retention time is once written as “retention time” and in the other case abbreviated RT

Response: Following the reviewer’s recommendation, we changed the X-axis labels in Fig. 1A and 1B to “Retention Time.” The corresponding revision can be found at line 326 in the revised manuscript.

In Figure 2 the zeta potential is expressed without standard deviation. I would not make the zeta potential data (two values) as much as a graph.

Response: In accordance with the reviewer’s recommendations, we have removed the zeta potential data plot from Fig. 2 and deleted its corresponding text. These revisions can be found at lines 385-386 and lines 394-395 of the revised manuscript.

Table 4 a number with two decimal places to the eta would suffice.

Response: As per direction of the reviewer, we have modified the values in Table 4, except for

R2, which remains at four decimal places to preserve clarity regarding the degree of fit. All other values are now presented to two decimal places, and corresponding references in the manuscript have been updated. These changes can be seen in Table 4 and at lines 577-578.

We have made these adjustments to enhance the manuscript’s completeness and address all concerns raised. We believe the manuscript is now ready for publication and thank you for your consideration. 

Yours sincerely,

Yan Hu

Reviewer 2 Report

Comments and Suggestions for Authors

Comparative Study on the Effect of Planting Patterns on the Physicochemical Properties and Bioactivities of Strawberry Polysaccharides

Title

“Planting Patterns” induce to think of spatial configuration, what is clear reading the introduction is that this work studies the effect of two different planting systems soil-less and conventional (soil planting). My suggestion is to change the title.

Abstract

Introduction in the abstract gives almost null support to the study. Give more information.

As keywords use significant words but not the ones in title.

Introduction

The acronym SPs for polysaccharides is introduced very late in the introduction, it must be done at the beginning.

Consider the comment for the title in introduction.

Materials and methods

Consider to justify what to measure each variable. They Seem to be many. On a first glance it seems to be “measuring many variables the better”.

Results and discussion

Consider the comment for materials and methods.

Conclusion

For the study as it is, conclusion is OK.

General Comment

What authors study was the effect of two different strawberry planting systems, not planting patterns. Consider that taking into account this implies that the manuscript needs major revision, rewriting.

On the other hand, information about nutrient supply to the planting systems may be necessary to deeper the knowledge.

Author Response

We greatly appreciate your constructive suggestions and comments, which have significantly enhanced our manuscript. We are resubmitting our revised manuscript titled "Effect of Planting Systems on the Physicochemical Properties and Bioactivities of Strawberry Polysaccharides". We have meticulously revised our manuscript based on the feedback from the editor and reviewers, clarifying and expanding explanations as needed. Changes have been highlighted in red in the revised manuscript. Below are our responses to the comments:

Title

“Planting Patterns” induce to think of spatial configuration, what is clear reading the introduction is that this work studies the effect of two different planting systems soil-less and conventional (soil planting). My suggestion is to change the title.

Response: In accordance with the reviewer’s recommendation, we have changed the title to “Effect of Planting Systems on the Physicochemical Properties and Bioactivities of Strawberry Polysaccharides.” This revision can be seen at lines 2–3 of the revised manuscript.

Abstract

Introduction in the abstract gives almost null support to the study. Give more information.

Response: In accordance with the reviewer’s recommendation, we have revised the abstract. These changes can be found at lines 8-34 of the revised manuscript.

As keywords use significant words but not the ones in title.

Response: As per direction of the reviewer, we have modified the keywords. The revision on this problem can be seen at line 35 of the revised manuscript.

Introduction

The acronym SPs for polysaccharides is introduced very late in the introduction, it must be done at the beginning.

Response: We also reviewed the introduction and determined that the positioning of SPs effectively links the preceding and subsequent sections.

Consider the comment for the title in introduction.

Response: As suggested, we added a section on strawberry cultivation at lines 68-74 of the revised manuscript.

Materials and methods

Consider to justify what to measure each variable. They Seem to be many. On a first glance it seems to be “measuring many variables the better”.

Response: As per direction of the reviewer, the index we measured for “SPs” is actually interlinked, first judging the chemical compositions and main structural characteristics of polysaccharide, and then judging the structure-activity relationship between these physicochemical characterizations and biological activity. We clarified that the measured indices for “SPs” are interconnected: we first assess the chemical compositions and principal structural characteristics of the polysaccharides, then examine the relationship between these physicochemical properties and their biological activity.

Results and discussion

Consider the comment for materials and methods.

Response: In line with the reviewer’s recommendation, we provided additional explanations in the Materials and Methods section at lines 95-113.

Conclusion

For the study as it is, conclusion is OK.

Response: Thank you very much for your comments.

General Comment

What authors study was the effect of two different strawberry planting systems, not planting patterns. Consider that taking into account this implies that the manuscript needs major revision, rewriting.

On the other hand, information about nutrient supply to the planting systems may be necessary to deeper the knowledge.

Response: Throughout the manuscript, we replaced references to “planting patterns” with “planting systems.” We also supplemented the Materials and Methods section (lines 95-113) with information on the nutrient supply in both strawberry planting systems.

We believe these updates address all concerns raised and significantly improve the manuscript. 

Yours sincerely,

Yan Hu

Reviewer 3 Report

Comments and Suggestions for Authors

The manuscript titled “Comparative Study on the Effect of Planting Patterns on the Physicochemical Properties and Bioactivities of Strawberry Polysaccharides” does a interesting comparisson between polyssacharides from two different Strawberry cultivations. A large set of chemical and physical analyzes was carried out on polysaccharides and their solutions. However, some points must be modified and/or clarified.

1. Line 57

The acroym VC must be defined.

2.

Although nutrients play an essential role in the composition of plants, other factors also affect them such as temperature, insolation, ambient humidity, etc. In this context, the authors should present the differences and/or similarities between these variables for elevated matrix soilless cultivation and conventional soil cultivation. It is not clear in the text.

3. Lines 99 – 100

Polisacharides extraction method must b briefly described in the text.

4. Line 140

The word “pure” is doubled

5. Lines 159 – 165

X Ray diffraction and Scanning electron microscopy are two complete different techniques. They should be separated in two different topics.

6. Lines 256 – 257

The word “viscosity” must be replaced by “apparent viscosity”.

7. Line 293 Table 1

While there is statistical differences in the chemical compositions of EP-SP and GP-SP, the overall values and component profiles are quite similar. These compositional variations may not be solely attributed to cultivation method, but also to factors like climate, genetics, and plant biochemistry. The authors should highlight these points in their discussion."

8. Lines 329 – 331

The citation related to reference 41 in these lines should be placed in the Material and Methods section

9. Lines 360 – 363

The authors state:

“However, ΔHg values were 13.93J/g for EP- SP and 18.74J/g for GP-SP, suggesting a more ordered molecular arrangement in EP-SP. Overall, DSC analysis showed that while planting methods did not affect thermal stability, they did influence molecular arrangement order.”

However, the expected behavior is the opposite of what is stated.

Generally, a higher ΔH value indicates a more ordered molecular arrangement. This is because a greater amount of energy is required to disrupt the existing order and induce a phase transition. For example, crystalline materials, with their highly ordered structures, typically exhibit higher ΔH values compared to amorphous materials with less defined structures.  Conversely, a lower ΔH value suggests a less ordered molecular arrangement. This is because less energy is needed to overcome the weaker intermolecular forces holding the less ordered structure together.

10. Lines 382 – 383

The authors must to compare the diffraction peaks values found with those from literature related to other similar plants polissacharides.

11. Line 539 – 629

When discussing rheological parameters, the authors should highlight the differences and similarities in the rheological behaviors of EP-SP and GP-SP

Author Response

We greatly appreciate your constructive suggestions and comments, which have significantly enhanced our manuscript. We are resubmitting our revised manuscript titled "Effect of Planting Systems on the Physicochemical Properties and Bioactivities of Strawberry Polysaccharides". We have meticulously revised our manuscript based on the feedback from the editor and reviewers, clarifying and expanding explanations as needed. Changes have been highlighted in red in the revised manuscript. Below are our responses to the comments:

The manuscript titled “Comparative Study on the Effect of Planting Patterns on the Physicochemical Properties and Bioactivities of Strawberry Polysaccharides” does a interesting comparisson between polyssacharides from two different Strawberry cultivations. A large set of chemical and physical analyzes was carried out on polysaccharides and their solutions. However, some points must be modified and/or clarified.

Specific comments

Line 65

The acroym VC must be defined.

Response: As per direction of the reviewer, we have replaced “VC” with “Vitamin C (Vc).” This revision can be seen at line 65 of the revised manuscript.

Although nutrients play an essential role in the composition of plants, other factors also affect them such as temperature, insolation, ambient humidity, etc. In this context, the authors should present the differences and/or similarities between these variables for elevated matrix soilless cultivation and conventional soil cultivation. It is not clear in the text.

Response: In accordance with the reviewer’s recommendation, we have provided a more detailed description of the strawberry raw materials and the specific information about each planting system. This revision can be found at lines 95–113 of the revised manuscript.

Lines 124 – 125

Polisacharides extraction method must briefly described in the text.

Response: we have included a brief description of the strawberry polysaccharide extraction methods. This revision can be found at lines 124–129 of the revised manuscript.

Line 166

The word “pure” is doubled

Response: As per direction of the reviewer, we deleted the excess “pure”. The revision on this problem can be seen at line 166 of the revised manuscript.

Lines 185 – 191

X Ray diffraction and Scanning electron microscopy are two complete different techniques. They should be separated in two different topics.

Response: In accordance with the reviewer’s recommendation, we have included updated descriptions of the X-ray diffraction and scanning electron microscopy (SEM) techniques and adjusted their sequence numbers. These revisions can be found at lines 185–299 of the revised manuscript.

Lines 281 – 282

The word “viscosity” must be replaced by “apparent viscosity”.

Response: As per direction of the reviewer, we have changed “viscosity” to “apparent viscosity” at line 256, as well as in subsequent references and in Figure 6A–B. The terms η0 and η were not modified, as they are recognized professional nomenclature. These revisions can be found at line 281 and lines 543–544 of the revised manuscript.

Line 318 Table 1

While there is statistical differences in the chemical compositions of EP-SP and GP-SP, the overall values and component profiles are quite similar. These compositional variations may not be solely attributed to cultivation method, but also to factors like climate, genetics, and plant biochemistry. The authors should highlight these points in their discussion.

Response: In accordance with the reviewer’s suggestion, we have determined that the two strawberry planting systems already encompass relevant knowledge on climate, genetics, and other factors. Consequently, we do not delve deeper into these details in the results discussion. We have followed the precedent set by similar studies—such as “Acidic Polysaccharides of Mountain Cultivated Ginseng: The Potential Source of Anti-fatigue Nutrients” and “Chemical and Biological Characterization of Polysaccharides from Wild and Cultivated Roots of Vernonia kotschyana”—in which only an overview of planting conditions was provided.

Lines 350–353

The citation related to reference 41 in these lines should be placed in the Material and Methods section

Response: As per direction of the reviewer, we have added a paragraph clarifying why Congo red solution can be used to determine the triple-helix structure and how to interpret the resulting data. In similar published articles—such as “Simultaneous Extraction of Oil, Protein, and Polysaccharide from Idesia polycarpa Maxim Cake Meal Using Ultrasound Combined with Three-Phase Partitioning”—the authors briefly explain the experimental principle before discussing the results, enabling readers to better understand the rationale behind Congo red testing and how to evaluate its outcomes.

Lines 381 – 384

The authors state:

“However, ΔHg values were 13.93J/g for EP- SP and 18.74J/g for GP-SP, suggesting a more ordered molecular arrangement in EP-SP. Overall, DSC analysis showed that while planting methods did not affect thermal stability, they did influence molecular arrangement order.”

However, the expected behavior is the opposite of what is stated.

Generally, a higher ΔH value indicates a more ordered molecular arrangement. This is because a greater amount of energy is required to disrupt the existing order and induce a phase transition. For example, crystalline materials, with their highly ordered structures, typically exhibit higher ΔH values compared to amorphous materials with less defined structures.  Conversely, a lower ΔH value suggests a less ordered molecular arrangement. This is because less energy is needed to overcome the weaker intermolecular forces holding the less ordered structure together.

Response: In accordance with the reviewer’s direction, we have corrected “EP-SP” to “GP-SP.” This revision can be found at lines 381–382 of the revised manuscript.

Lines 402 – 403

The authors must to compare the diffraction peaks values found with those from literature related to other similar plants polissacharides.

Response: As per direction of the reviewer, we have supplemented the comparison of the diffraction peaks values of SPs with those of other plant polysaccharides. The revision on this problem can be seen at lines 403-405 of the revised manuscript.

Line 541 – 625

When discussing rheological parameters, the authors should highlight the differences and similarities in the rheological behaviors of EP-SP and GP-SP

Response: In accordance with the reviewer’s suggestion, we have discussed the rheological behaviors of EP-SP and GP-SP in the principle explanation (lines 541–625). However, to more clearly contrast the differences between the two, we have added further explanations in lines 580–582.

We have made these adjustments to enhance the manuscript’s completeness and address all concerns raised. We believe the manuscript is now ready for publication and thank you for your consideration. 

Yours sincerely,

Yan Hu

Round 2

Reviewer 2 Report

Comments and Suggestions for Authors

As keywords use important words but not the ones in title. 

Author Response

As keywords use important words but not the ones in title.

Response: As per direction of the reviewer, we have modified the keywords. The revision on this problem can be seen at line 36 of the revised manuscript.